# Reinforcement learning for one-shot DAG scheduling with comparability identification and dense reward

Xumai Qi[1,2]      Dongdong Zhang[1]*      Taotao Liu[1]      Hongcheng Wang[1]

[1]School of Computer Science and Technology, Tongji University
[2]Shanghai Key Laboratory of Urban Renewal and Spatial Optimization Technology, CAUP, Tongji University

## Abstract

In recent years, many studies proposed to generate solutions for Directed Acyclic Graph (DAG) scheduling problem in one shot by combining reinforcement learning and list scheduling heuristic. However, these existing methods suffer from biased estimation of sampling probabilities and inefficient guidance in training, due to redundant comparisons among node priorities and the sparse reward challenge. To address these issues, we analyze of the limitations of these existing methods, and propose a novel one-shot DAG scheduling method with comparability identification and dense reward signal, based on the policy gradient framework. In our method, a comparable antichain identification mechanism is proposed to eliminate the problem of redundant nodewise priority comparison. We also propose a dense reward signal for node level decision-making optimization in training, effectively addressing the sparse reward challenge. The experimental results show that the proposed method can yield superior results of scheduling objectives compared to other learning-based DAG scheduling methods.

## 1 Introduction

The Directed Acyclic Graph (DAG) scheduling problem is a class of NP-hard [Kan, 2012] Combinatorial Optimization Problems (COP). DAG scheduling problem arises in the emerging cloud manufacturing and distributed computing, involving production and computation workflow scheduling. The jobs, operations, or tasks are modeled into nodes of DAG, and the directed edges represent the precedence constraints among them. The goal is to achieve the best possible performance by determining an optimal node execution order and allocating resources accordingly.

DAG scheduling has attracted extensive research attention, with many heuristic, metaheuristic and Reinforcement Learning (RL) methods proposed to solve it. Among them, RL has already shown promise in solving generic COPs [Kwon et al., 2020, Hottung et al., 2022, Choo et al., 2022, Zhang et al., 2023, Wang et al., 2025]. Specifically for DAG scheduling, early studies tended construct the solution (i.e., the order of node execution) incrementally [Mao et al., 2019, Zhou et al., 2022, Song et al., 2023]. They utilized graph neural networks (GNN) to guide the policy network to select the optimal schedulable node at each step, until the complete solution is constructed.

Considering that the incremental solution construction methods are computationally consuming and less efficient in training, some studies proposed RL-based one-shot DAG scheduling, which is motivated by list scheduling [Lee et al., 2020, 2021, Jeon et al., 2023]. They attempted to enable network to infer the entire solution through a single forward propagation process. List scheduling is a classic and effective heuristic rule for DAG scheduling. As shown in Fig.1a, they generate a full schedule of the DAG by leveraging GNN to assign a fixed priority list for all task nodes in one single step, and iteratively learn to optimize the network according to the objective cost as the feedback

---

*Corresponding Author

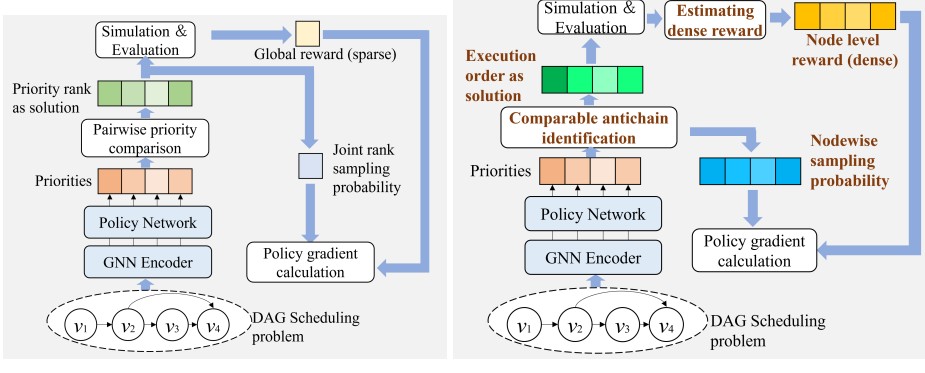

(a) Existing method.      (b) Our proposed method.

Figure 1: The comparison of existing one-shot DAG scheduling method and ours.

reward. Compared to the incremental solution construction methods, such one-shot scheduling approaches generally yields better results and requires less computation.

However, as we will analyze in detail in section 3, the existing RL-based one-shot scheduling approaches involve excessive node priority comparisons and sparse reward signal challenge, leading to unstable training and suboptimal results. They unnecessarily compare node priorities in pairwise, rather than accurately representing the distribution of execution order, leading to biased policy sampling probability and high variance in policy gradient estimation. Moreover, the sparse reward problem remains a challenge in the one-shot approaches. The agent has to optimize the entire priority list with a limited amount of global reward signal.

To address these issues, as shown in Fig. 1b, based on our analysis, we propose a novel RL-based one-shot DAG scheduling method with comparability identification and dense reward signal. (We define *comparable* to distinguish the task pairs that would influence the priority-induced execution orders.) For the redundant node priority comparison problem in the existing RL-based one-shot scheduling approaches, we propose a comparable antichain identification mechanism to eliminate the problem of redundant nodewise priority comparison. In this way, we can derive the accurate formulation of the nodewise sampling probability corresponding to the generated solution, thereby avoiding inaccurate probability estimation and improving training stability. Besides, compared to the existing methods suffering from sparse reward challenge, we design a dense reward for one-shot RL-based DAG scheduling methods to guide node-level optimization, enabling the scheduling results to closely converge to the optimum. We modify the key components in the existing RL-based one-shot DAG scheduling method, while retaining the list-based scheduling components that have proven effective. **The key contributions of this study are as follows:**

1. We analyze the limitations of existing RL-based one-shot methods in DAG scheduling, including the redundant nodewise priority comparison problem and the sparse reward challenge.

2. Based on the analysis, we propose a novel RL-based one-shot solution generation method for DAG scheduling. In our method, a comparable antichain identification mechanism is proposed to eliminate the problem of redundant nodewise priority comparison. We also propose a dense reward signal for node level decision-making optimization in training, effectively addressing the sparse reward challenge.

3. Comprehensive comparative and ablation experiments conducted on various DAG scheduling tasks demonstrate the superiority of our method in terms of solution quality.

## 2 Preliminaries

**DAG scheduling.** We define a DAG $G = (V, E)$ , where the node set $V = \{v_1, v_2, ..., v_n\}$ represents the tasks in the real-world DAG scheduling problem, and the directed edge set $E \subseteq V \times V$ denotes the precedence constraints among the tasks. If $(v_i, v_j) \in E$, task $v_j$ cannot start until $v_i$ is completed. That is, a task node can be ready only after all its predecessors is finished. Each task $v_i$ is

typically associated with a processing time $d_i$, and $d_i$ might be related to the processor where the task is allocated. The scheduler needs to determine a feasible task node execution order, and processor allocation if in multi-processor environments. The goal is to optimize the target cost metric evaluated after completing the DAG execution, such as makespan.

**List scheduling heuristic for DAG scheduling.** In list scheduling, according to a certain ranking rule $\sigma : \{1, 2, ..., n\} \rightarrow \{1, 2, ..., n\}$, a list of priority rank $Rank(V) = [v_{\sigma(1)}, v_{\sigma(2)}, ..., v_{\sigma(n)}]$ for each node in $V$ is generated. At each step, the ready task that has the highest priority rank is selected to be scheduled .

**Policy gradient.** In a typical policy gradient framework, the gradient to update the network parameters $\theta$ is formulated as Equation (1). $p_\theta(a_t|s_t)$ is the probability for the RL network with parameters $\theta$ to sample $a_t$ according to the current $s_t$. $R_t$ is the cumulative single-step reward $r$ starting from $t$.

$$\nabla_\theta J(\theta) = \mathbb{E}_{p_\theta} \left[ \sum_{t=1}^{T} \nabla_\theta \log p_\theta(a_t|s_t) \cdot R_t \right] \tag{1}$$

**One-shot DAG scheduling.** The existing approaches (e.g.,Jeon et al. [2023]) follow the paradigm of list scheduling and policy gradient. Instead of making stepwise decisions, they define the action space as the priority rank of all task nodes in the DAG. As shown in Fig. 1a, the DAG scheduling problem instance is fed into the RL network $\theta$, which consists a GNN encoder and a policy network, outputting logits for each node. The logits are treated as the priority values of each task node. The corresponding ranking rule $\sigma_\theta$ is to perform a descending sort on the logits of each task node. In this way, the priority rank $Rank(V; \theta) = [v_{\sigma_\theta(1)}, v_{\sigma_\theta(2)}, ..., v_{\sigma_\theta(n)}]$ can be sampled. Sampling such a descending sort can essentially be interpreted as iteratively selecting the maximum element from the remaining unselected ones, which involves pairwise priority comparison of all task nodes. At each selection step, the sampling probability can be calculated by performing softmax operation over the current candidates $V - \{v_{\sigma_\theta(1)}, ..., v_{\sigma_\theta(t-1)}\}$, as shown in Equation (2). The probability of the entire rank can be calculated as Equation (3). In one-shot DAG scheduling, Equation (1) can be modified to Equation (4). where $C(G, Rank(V; \theta))$ is the value of optimization goal when applying the sampled priority rank $Rank(V; \theta)$ on $G$.

$$p_\theta(v_{\sigma_\theta(t)}|[v_{\sigma_\theta(1)}, ..., v_{\sigma_\theta(t-1)}]) = \frac{\exp(\text{logits}_\theta(v_{\sigma_\theta(t)}))}{\sum_{v \in V - \{v_{\sigma_\theta(1)}, ..., v_{\sigma_\theta(t-1)}\}} \exp(\text{logits}_\theta(v))} \tag{2}$$

$$P(Rank(V; \theta)) = \prod_{t=1}^{n} \frac{\exp(\text{logits}_\theta(v_{\sigma_\theta(t)}))}{\sum_{v \in V - \{v_{\sigma_\theta(1)}, ..., v_{\sigma_\theta(t-1)}\}} \exp(\text{logits}_\theta(v))} \tag{3}$$

$$\nabla_\theta J(\theta) = \mathbb{E}_{P(Rank(V; \theta))} \left[ \nabla_\theta \log P(Rank(V; \theta)) \cdot C(G, Rank(V; \theta)) \right] \tag{4}$$

## 3 Analysis of the one-shot schedule approaches

### 3.1 Redundant priority comparison and biased sampling probability estimation

We observe that the current methods' action space is inconsistent with the actual scheduling solutions. The existing methods directly regard the output priority rank $Rank(V; \theta)$ as the action of RL. However, multiple priority ranks may correspond to a same node execution sequence. Some heuristic-based priority generating approaches, such as Topcuoglu et al. [2002], Djigal et al. [2021], are designed to ensure that the obtained node priority rank is also the node execution order, but the logits output by a neural network cannot inherently guarantee such consistency. (For example, in Fig. 2, both $Rank(V; \theta_1)$ and $Rank(V; \theta_2)$ result in the same node execution order for the graph on the left.) If multiple priority ranks $Rank_1, Rank_2, ...$ correspond to a same node execution sequence $Solution(G) = [v_{\pi(1)}, v_{\pi(2)}, ..., v_{\pi(n)}]$, the total probability of sampling that solution should be Equation (5). So, if only a single rank list's probability $P(Rank_j)$ is used to approximate

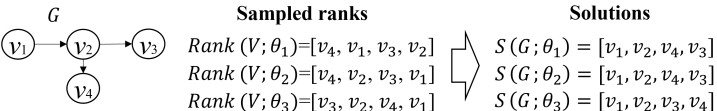

Figure 2: The examples of different priority rank that which can be mapped to a same node execution order, and the impact of inaccurate probability estimation on training.

$P(Solution(G))$, this introduces underestimation in probability. This increases entropy and the variance of the policy gradient.

$$P(Solution(G)) = P(Rank_1 \cup Rank_2 \cup \dots) \tag{5}$$

Such inconsistency causes redundant comparison. Given a one-shot RL network $\theta$, the solution $Solution(G; \theta)$ is a priority topological sort of $G$ based on $Rank(V; \theta)$. This sort is constrained by both the priority ranks and the directed edges $E$. $E$ are hard constraints, while the priority ranks only serve as soft constraints. If there exists a path from $u$ to $v$, then $u$ is guaranteed to precede $v$ in any valid $Solution(G)$, regardless of their priority. The priority comparison of nodes located on the same path is unnecessary. But when constructing a priority rank, existing methods iteratively select the node with the maximum priority by comparing it with all unselected nodes, including the nodes that are on the same path of the selected node. These comparisons are thus redundant or invalid, as they do not affect the resulting topological order. Treating them as candidate actions introduces irrelevant logits into the softmax distribution, distorting the sampling probability.

Redundant priority comparison and biased sampling probability estimation may lead to unstable training and suboptimal results. Under pairwise priority comparison, the model may tend to equally optimize the pairwise ordering of all node pairs, rather than focusing on those node pairs that actually influence the final schedule. The model might spend unnecessary effort on optimizing the rank of nodes on the same paths. As a result, the training can become less efficient. Taking Fig. 2 for example, the final node execution order is solely determined by the relative priorities between $v_3$ and $v_4$. If the optimization objective cost $C(G, Rank(V; \theta_1))$ obtained from $Rank(V; \theta_1)$ achieves better than $C(G, Rank(V; \theta_3))$, the model might mistakenly attribute the improvement to $v_2$'s being higher-ranked than $v_1$. However, since $v_1$ and $v_2$ lie on the same path, their relative priorities have no influence on the result. Similarly, the model may spend unnecessary effort optimizing other node pairs that are irrelevant to the outcome, such as $< v_1, v_3 >$, $< v_1, v_4 >$.

### 3.2 Sparse reward challenge

In addition to the redundant priority comparison, we also need to address the sparse reward challenge in the one-shot solution generation methods. Specifically, the global reward is generated only once in the whole decision-making process, which cannot effectively reflect the specific contribution of each local decision (the priority assignment of each task node) to the final result, thus seriously limiting the effective guidance of policy gradient algorithm for network parameters. This sparsity not only makes the training, but also may cause the model to get trapped in a local optimum. Therefore, designing a denser and more decomposable reward signal to guide local decision-making is a key issue to improve the training efficiency and stability of one-shot method.

In conclusion, the biased sampling probability estimation caused by redundant priority comparison, together with the sparse reward challenge, would constrain the performance of RL-based one-shot DAG scheduling. We will address these two issues in the next section.

## 4 Proposed method

### 4.1 Motivation

To address the issue of redundant priority comparison, according to the analysis in the previous section, we need to distinguish the nodes that has necessity to compare. Removing those invalid nodes in each step aligns the actual sampling distribution closer to the true $P(Solution(G))$. It can produce lower-entropy distributions and reduced policy gradient variance, which is beneficial to the

convergence for RL model [Huang and Ontañón, 2020]. Clearly, there is no need to compare the priorities of node pairs lying on the same path. Only mutually unreachable node pairs (i.e., those belonging to the same antichain) require comparisons. However, since a node may appear in multiple different antichains, it is necessary to determine which antichain comparisons are relevant. For example, assume that $\{v, u_1\}$ and $\{v, u_2\}$ are respectively two antichains in a graph, with a path between $u_1$ and $u_2$. If $v$ is prior to $u_1$ in $Rank(V)$, $v$ is guaranteed to precede $u_2$ in $Solution(G)$, no matter which one in $v_1$ and $v_2$ is prior. This makes the priority comparison between $u_1$ and $v_2$ unnecessary. But if $u_1$ is prior to $v$, the comparison between $v_1$ and $v_2$ remains essential to determine the final order of $v$ and $u_2$ in $Solution(G)$.

It is challenging to define a meaningful stepwise reward for DAG scheduling problems. However, it is still practical to estimate the cumulative return $R_t$ in one-shot scheduling. The cumulative reward at each step reflects the difference between the contribution of the current/local decision and the global objective. Since the objective value associated with each task node can be obtained during the simulation-based evaluation of the overall scheduling result, the cumulative reward can be estimated accordingly.

In this section, we propose a one-shot solution generation method for DAG scheduling based on policy gradient algorithm framework that can address the above-mentioned problems. Specifically, we first propose a comparable antichain identification method for each node during the process of priority topological sorting. This enables us to compute nodewise sampling probabilities and obtain an execution sequence as the solution. The simulated or evaluated cost of each node in the resulting schedule is then compared with a heuristic-derived advantage baseline to generate node level dense reward signals. These dense rewards, along with the sampling probabilities, are used to calculated the policy gradient. The overview figure of the proposed method is illustrated in Appendix A.

Our framework adopts a GNN-based encoder and a multi-layer perceptron policy (MLP) policy network to handle the DAG scheduling problem instance to produce priority logits list for all nodes, following the common setup in prior work. The encoder is implemented using the Graphormer [Ying et al., 2021], whose effectiveness in encoding directed graph data has been widely validated. Check Appendix C for more details about the implementation about our method.

## 4.2 Comparable antichain identification

We define a pair of nodes as comparable if their relative priorities affect the resulting execution order, as formally stated in **Definition 1**. Based on this definition, we further define the comparable antichain of a node in **Definition 2**.

**Definition 1 (Comparable node pair):** Given task nodes $v$ and $u$ in a DAG scheduling problem, a priority rank $Rank(V)$, and the node execution sequence $Solution(G)$ derived from $Rank(V)$. Without loss of generality, assume that $u$ is higher-ranked than $v$ (i.e., $u$ is before $v$ in $Rank(V)$). Remove and reinsert both $u$ and $v$ into any position in $Rank(V)$ while guaranteeing $u$ is now after $v$, obtaining a $Rank'(V)$. If, for any $Rank'(V)$, the relative ordering of $u$ and $v$ in $Solution'(G)$ derived from $Rank'(V)$ is always the reserve of that in $Solution(G)$, we say that $< u, v >$ is **comparable** in $Rank(V)$.

**Definition 2 (Comparable antichain):** The set that contains $v$ and the comparable nodes of $v$ in $Rank(V)$.

We propose a straightforward comparable antichain identification method based on the priority topological sorting process, which aims to eliminate redundant priority comparisons between incomparable nodes and accurately reflect the nodewise sampling probability corresponding to the sampled specific node execution sequence. Recalling the process of priority topological sorting: given a DAG $G$, iteratively remove the already sorted nodes from $G$ in each step $t$, obtaining a subgraph $G_t$. Then in $G_t$, identify the set of nodes with 0 in-degree $V_t^{in=0}$, and select the node $v_{\pi(t)}$ with hightest priority rank from the 0-in-degree set $V_t^{in=0}$, and append it to $S_t(G)$.

Clearly, the 0-in-degree set $V_t^{in=0}$ in each step forms an antichain of $G$, since the node removal operations do not remove edges between the remaining nodes and the nodes with 0 in-degree is guaranteed to be mutually unreachable. According to **lemma 1** and **lemma 2**, we can track the comparable antichain of each $v_{\pi(t)}$ by merely using $V_t^{in=0}$.

**Lemma 1:** $V_t^{in=0}$ is a comparable antichain of $v_{\pi(t)}$.

**Proof:** By construction, $v_{\pi(t)}$ is higher-ranked than any $u \in V_t^{in=0}$, so $v_{\pi(t)}$ appears before each such $u$ in the resulting $Solution(G)$. Now suppose to re-insert the priority rank so that $u$ is higher-ranked to $v_{\pi(t)}$. Then either $u$ is selected before step $t$, or $u$ is select instead of $v_{\pi(t)}$ at $t$. In both cases, $u$ would now be before $v_{\pi(t)}$ in the resulting $Solution'(G)$. Therefore, $V_t^{in=0}$ forms a comparable antichain of $v_{\pi(t)}$.

**Lemma 2:** For any $u \in V_t^{in=0}$, none of its successors is comparable antichain with $v_{\pi(t)}$.

**Proof:** By construction, $v_{\pi(t)}$ is higher-ranked than any $u \in V_t^{in=0}$ and $v_{\pi(t)}$ appears before each $u$ in $Solution(G)$. $u'$, a successor of $u$, is guaranteed to appear after $u$ in $Solution(G)$ due to the precedence constraints, it must also appear after $v_{\pi(t)}$ in any valid $Solution'(G)$ by re-inserting $v_{\pi(t)}$ and $u'$ in the priority rank. Thus, regardless of how the relative priorities between $v_{\pi(t)}$ and $u'$ are changed, their positions in $Solution'(G)$ remain fixed.

In this way, the nodewise sampling probability corresponding to $Solution(G)$, together with the probability of $Solution(G)$ can be precisely obtained. The procedure of "selecting $v_{\pi(t)}$ from $V_t^{in=0}$" can be interpreted as performing an argmax operation in $V_t^{in=0}$. Similar with Jeon et al. [2023], instead of performing argmax on the original logits of $V_t^{in=0}$, we use the gumbel-perturbed logits[Kool et al., 2019] of $V_t^{in=0}$. Therefore, the sampled rank sequence becomes more aligned with the underlying probability distribution. Given the logits generated by the policy network in one shot, the softmax-based nodewise sampling probability can be calculated in Equation (6).

$$p_\theta(v_{\pi_\theta(t)}|S_{t-1}(G;\theta) = [v_{\pi_\theta(1)}, ..., v_{\pi_\theta(t-1)}]) = \frac{\exp(\text{logits}_\theta(v_{\pi_\theta(t)}))}{\sum_{v \in V_t^{in=0}} \exp(\text{logits}_\theta(v))} \qquad (6)$$

### 4.3 Dense reward

We further propose a dense reward signal for RL-based one-shot DAG scheduling, providing nodewise reward signal to guide the optimization of local decisions. For each scheduled node, we define its dense reward signal based on its distance to the final objective, thus approximating the node's value estimation. Such design also makes one-shot DAG scheduling more interpretable. Specifically, the nodewise decisions, sampling probabilities and dense reward signals can be regarded as an entire sampling trajectory in RL (like the trajectory by Monte Carlo sampling). This makes the interpretability of our methods closer to MDP-based incremental approaches than those one-shot methods with sparse reward signal signal $C(G, Solution(G;\theta))$ or $C(G, Rank(G;\theta))$.

Considering Equation (1), we regard the selection of the hightest priority node from $V_t^{in=0}$ at each step $t$ as the "action", instead of treating the generation of the whole priority rank or execution order as the single action. The sampling probability of each step can be calculated in Equation (6). Therefore, Equation (1) can be modified into Equation (7).

$$\nabla_\theta J(\theta) = \frac{1}{n} \sum_{t=1}^{n} \nabla_\theta \log p_\theta(v_{\pi_\theta(t)}|S_{t-1}(G;\theta)) \cdot R_t \qquad (7)$$

We estimate $R_t$ by Equation (8). Here, $C(S_t(G;\theta))$ is the objective cost function value corresponding to the current subsequence $S_t(G;\theta)$. For instance ,if the optimization goal is to minimizing the DAG's **makespan**, then $C(S_t(G;\theta))$ indicates the latest completion time among all scheduled nodes when $v_{\pi_\theta(t)}$ is finished. The value of $C(S_t(G;\theta))$ can be obtained during the simulation process when computing $C(Solution(G;\theta))$.

$$R_t = C(Solution(G;\theta)) - C(S_t(G;\theta)) \qquad (8)$$

We further consider introducing advantage baseline to $R_t$, in order to stabilize the training. The advantage baseline is computed by a heuristic algorithm (e.g., critical path method). Specifically, before training on DAG scheduling problem $G$, we schedule $G$ using the heuristic baseline, and record the objective cost value upon the completion of each node $v_i$, denoted as $C_h(v_i)$. Based on this, we can define the advantage function $A_t$ as shown in Equation (9). $A_t$ can be further normalized in batch-wise into $A_t^{\text{normalized}(j)}$. In this way, Equation (7) can be modified into Equation (10), where

$B$ is the batch size. The detailed procedure of calculating dense reward signal is demonstrated in Appendix D.

$$A_t = R_t - \mathbf{baseline} = C_h(v_{\pi_\theta(t)}) - C(S_t(G; \theta)) \tag{9}$$

$$\nabla_\theta J(\theta) = \frac{1}{B} \frac{1}{n} \sum_{j=1}^{B} \sum_{t=1}^{n} \nabla_\theta \log p_\theta(v_{\pi_{\theta^{(j)}}(t)} | S_{t-1}(G; \theta^{(j)})) \cdot A_t^{\text{normalized}(j)} \tag{10}$$

In practice, compared to sparse reward settings (where only the final cost is used), our dense reward reduces the variance of policy gradient estimates, because each node's policy update receives a targeted reward signal rather than sharing a single scalar. Also, it helps the model differentiate good and bad actions early in training.

## 5    Related work

RL has demonstrated significant potential in solving DAG scheduling problems in an end-to-end manner. A typical RL DAG scheduler consists of a GNN encoder extracting graph features, and a policy network making scheduling decisions [Mao et al., 2019, Zhou et al., 2022, Song et al., 2023]. There are already some GNN variants for general DAG-structured applications [Yu et al., 2019, Thost and Chen, 2021, Luo et al., 2023]. Some focused on modifying GNNs specially for DAG scheduling [Gagrani et al., 2022, Zhang et al., 2024]. The other studies paid attention to improving the decision-making, including modifying the policy network or action space. From the viewpoint of decision-making in RL-based DAG scheduling, existing literature can be categorized into **incremental solution construction**, **one-shot schedule generation**, and **edge-generation** approaches.

**Incremental solution construction.**    The agents follow Markov Decision Process (MDP) to select a candidate task node at each step, constructing the whole solution incrementally [Yang et al., 2019, Yu et al., 2023, Dong et al., 2023, Qi et al., 2024]. These methods faces two major limitations, which cause inefficient exploration and training. The first is the sparse reward challenge. Some works [Chen et al., 2023, Wang et al., 2025, Nasuta et al., 2024] tried to overcome this challenge only in specific application domains, rather than providing general solutions. Second, these methods require to re-encode the entire graph instance in each step, leading to excessive computation.

**One-shot schedule generation.**    Inspired by the list scheduling heuristic, some studies learn to generate DAG schedules in one shot. Lee et al. [2020, 2021] proposed to use sequence learning technique to generate a global fixed priority list for all task nodes in one single step. Jeon et al. [2023] proposed a node prioritization method based on the Gumbel-max-k trick [Kool et al., 2019], which enables efficient sampling of node priority values. One-shot scheduling avoids repeatedly extracting graph features and reduces the exploration space. However, the one-shot methods exhibit lower interpretability compared to incremental construction methods for not following MDP [Darvariu et al., 2024] . Moreover, when applied to large-scale instances, one-shot methods is still limited by sparse reward challenge.

**Edge generation.**    Some studies attempt to prioritize task nodes indirectly by modifying the DAG's topological structure, rather than directly generating solutions. Wang et al. [2021] and EGS proposed by Sun et al. [2024] leverages the property of DAG scheduling problem that a solution to remains valid after adding new edges. They iteratively generate edges using RL, aiming to improve the solution obtained by traditional heuristics. However, as we will later demonstrate, these approaches often suffer from an excessively large search space.

## 6    Experiments

In this section, we report the experimental results to evaluate both the individual contributions of our proposed modules and the overall performance of our method compared with existing baselines.

Most of the training, simulation and evaluation of the experiments are conducted on a machine with an Intel Gold 6226R CPU, 256 GB of RAM and NVIDIA RTX 3090 (24G) GPU.

**Benchmarks setup.** In order to emphasize that our method is a general method for DAG scheduling rather than being tied to a specific problem, we evaluate our method on three benchmarks with unique environment settings, reflecting the method's adaptability to diverse scenarios. These benchmarks are **Pegasus**, **TPC-H** , and a job shop scheduling problem (**JSSP**) benchmark generated by Zhang et al. [2020] . Pegasus is a scientific workflow scheduling tracing dataset with heterogeneous multiprocessor setting [Deelman et al., 2015]. We use earliest-finish-time greedy rule to allocate processors. The classic Heterogeneous Earliest Finish Time (HEFT) [Topcuoglu et al., 2002] algorithm is adopted as the advantage baseline for Pegasus benchmark in our experiments when estimating the dense reward. TPC-H represents the DAG workflow scheduling task under homogeneous resource environment. We use the TPC-H benchmark generated by Wang et al. [2021] in the experiments. Shortest First Time (SFT) is adopted as the heuristic baseline for TPC-H. JSSP is a special case of DAG scheduling under multi-machine settings. Shortest Processing Time (SPT) is adopted as the heuristic baseline for JSSP. Check Appendix B for more details about thees benchmarks. Evaluation metrics include the optimization objective, gap to heuristic baseline, and run time of the algorithm.

**Baselines.** Our approach is compared against the following baselines: (1) The heuristic algorithms used as advantage baselines in our method; (2) Jeon et al. [2023], a state-of-the-art RL-based one-shot DAG scheduling method; (3) **EGS** [Sun et al., 2024], a DAG scheduling approach based on edge generation; (4) **POMO-DAG**, our adapted implementation of the POMO [Kwon et al., 2020] for DAG scheduling, serving as a incremental solution construction method. See Appendix C for more details about these baselines. Additionally, we conduct ablation studies to assess the contribution of each proposed component in section 4, including: (5) **Ours without CAI**, our method without conducting comparable antichain identification (CAI); (6) **Ours without DR**, our method without estimating gradient with dense reward (DR), but sparse reward. The results are presented in Tab. 1, 2 and 3. More results on other workflows in the Pegasus benchmark is presented in Appendix F.

**Ablation studies.** From the perspective of scheduling performance in optimization objective, we observe that removing the comparable antichain identification module generally results in a larger degradation compared to removing the dense reward module. This suggests that redundant pairwise node priority comparisons are more detrimental to solution quality, while the dense reward acts more like a "fine-tuning". Note that the results of our full method on the TPC-H in Tab. 2 do not always outperform either the variant without comparable antichain identification or Jeon et al. [2023]'s one-shot approach. A possible explanation is that the DAGs in each problem of TPC-H are small and less interconnected. In this situation, the network may have difficulty learning the node priorities across the DAG, and since CAI has masked out many non-comparable nodes, this further increases the difficulty of learning in such cases. In terms of runtime, incorporating comparable antichain identification, which is based on the priority topological sorting process, slightly increases the overall run time. This is because priority topological sorting introduces additional computation with a time complexity of $O(n\log n + |E|)$, which is higher than ranking via argsort operation ($O(n\log n)$). Furthermore, priority topological sorting involves loops that, unlike argsort, cannot be parallelized using GPU-accelerated libraries. Nevertheless, the additional runtime overhead remains within an acceptable range. In summary, both comparable antichain identification and dense reward make contributions to the overall performance, with comparable antichain identification playing a more critical role.

Table 1: Experimental results on SIPHT dataset in Pegasus, with 5 different problem instance sizes (100, 200, 300, 400, and 1000 task nodes). The evaluation metrics include the optimization objective (Makespan), the gap relative to the heuristic baseline (HEFT), and the run time.

| Method | SIPHT-100 | | | SIPHT-200 | | | SIPHT-300 | | | SIPHT-400 | | | SIPHT-1000 | | |
|---|---|---|---|---|---|---|---|---|---|---|---|---|---|---|---|
| | MS | Gap/% | Time/s | MS | Gap/% | Time/s | MS | Gap/% | Time/s | MS | Gap/% | Time/s | MS | Gap/% | Time/s |
| HEFT (baseline) | 227.0 | - | 0.005 | 357.8 | - | 0.018 | 543.0 | - | 0.024 | 714.8 | - | 0.044 | 1821.4 | - | 0.196 |
| Jeon et al. [2023] | 218.5 | -3.74 | 0.07 | 352.2 | -1.57 | 0.15 | 550.6 | 1.40 | 0.26 | 712.7 | -0.29 | 0.43 | 1898.1 | 4.21 | 2.44 |
| POMO-DAG | 214.0 | -5.74 | 13.4 | 367.5 | 2.72 | 20.8 | 575.8 | 6.05 | 27.5 | 741.9 | 3.80 | 34.3 | 1875.1 | 2.95 | 50.6 |
| EGS | 200.6 | -11.63 | 1.02 | 346.3 | -3.21 | 2.35 | 542.8 | -0.04 | 4.04 | 710.2 | -0.64 | 10.2 | 1821.0 | -0.02 | 64.5 |
| Ours | **196.9** | **-13.3** | 0.61 | **338.4** | **-5.42** | 0.97 | **541.6** | **-0.25** | 1.17 | **708.3** | **-0.91** | 1.22 | 1819.2 | -0.13 | 2.59 |
| Ours w/o DR | 213.2 | -6.07 | 0.35 | 345.6 | -3.41 | 0.67 | 542.2 | -0.14 | 1.38 | 710.3 | -0.63 | 1.88 | **1818.9** | **-0.13** | 2.83 |
| Ours w/o CAI | 214.4 | -5.55 | 0.11 | 345.8 | -3.07 | 0.21 | 542.5 | -0.09 | 0.34 | 710.4 | -0.62 | 0.46 | 1867.7 | 2.54 | 2.22 |

Table 2: Experimental results on TPC-H benchmark, with 3 different problem instance sizes (50, 100 and 150 sub-DAGs). The evaluation metrics include the optimization objective makespan (MS), the gap relative to the heuristic baseline (STF), and the run time.

| Method | TPC-H 50 | | | TPC-H 100 | | | TPC-H 150 | | |
|---|---|---|---|---|---|---|---|---|---|
| | MS | Gap/% | Time/s | MS | Gap/% | Time/s | MS | Gap/% | Time/s |
| STF (baseline) | 24.97 | - | 0.010 | 42.85 | - | 0.027 | 69.76 | - | 0.038 |
| Jeon et al. [2023] | 23.73 | -4.95 | 0.23 | 41.22 | -3.81 | 0.44 | 74.02 | 6.11 | 0.76 |
| POMO-DAG | 46.90 | 87.8 | 24.3 | 90.30 | 110.74 | 27.4 | 141.90 | 103.43 | 33.2 |
| EGS | 24.58 | -1.55 | 10.8 | 42.18 | -1.56 | 51.2 | 68.99 | -1.10 | 156.8 |
| Ours | **20.49** | **-17.73** | 0.51 | **39.22** | **-8.47** | 0.82 | 73.47 | 5.32 | 1.49 |
| Ours w/o DR | 24.10 | -3.47 | 0.59 | 39.68 | -7.40 | 0.86 | 70.14 | 0.54 | 1.19 |
| Ours w/o CAI | 22.47 | -10.00 | 0.23 | 42.96 | 0.16 | 0.43 | **67.91** | **-2.65** | 0.78 |

Table 3: Experimental results on JSSP.

| Method | JSSP 20*10 | | | JSSP 20*20 | | | JSSP 30*10 | | | JSSP 30*20 | | |
|---|---|---|---|---|---|---|---|---|---|---|---|---|
| | MS | Gap/% | Time/s | MS | Gap/% | Time/s | MS | Gap/% | Time/s | MS | Gap/% | Time/s |
| SPT (Baseline) | 516.7 | - | <0.001 | 1096.2 | - | <0.001 | 845.9 | - | <0.001 | 1692.0 | - | <0.002 |
| Jeon et al. [2023] | 445.2 | -13.84 | 0.12 | 964.0 | -12.06 | 0.15 | 735.6 | -13.04 | 0.24 | 1548.6 | -8.48 | 0.15 |
| POMO-DAG | **341.6** | **-33.88** | 3.0 | 936.3 | -14.59 | 5.3 | 971.3 | -6.45 | 4.2 | 1458.0 | -13.83 | 6.7 |
| EGS | 465.9 | -9.83 | 1.8 | 1034.5 | -5.63 | 3.5 | 837.5 | -0.99 | 6.7 | 1604.1 | -5.20 | 17.0 |
| Ours | 397.3 | -23.11 | 0.32 | **813.8** | **-25.76** | 0.41 | **571.3** | **-32.46** | 0.37 | **1426.0** | **-15.72** | 0.45 |
| Ours w/o DR | 394.7 | -23.61 | 0.36 | 928.6 | -15.29 | 0.41 | 661.1 | -21.85 | 0.32 | 1481.9 | -12.42 | 0.44 |
| Ours w/o CAI | 436.8 | -15.46 | 0.12 | 939.2 | -14.32 | 0.16 | 718.2 | -15.10 | 0.12 | 1505.3 | -11.03 | 0.14 |

**Comparison experiments.** Our method achieves better performance than Jeon et al. [2023] in most cases. But it incurs slightly longer runtime, as explained in the ablation study. When compared with POMO-DAG, our method consistently yields better results in most scenarios. In a few small-scale cases such as JSSP-20-10 in Tab. 3, our method performs slightly worse. We attribute this to the simple disjunctive DAG structure of JSSP, where the advantage of comparability identification becomes less significant. Nonetheless, our approach demonstrates a substantial advantage in run time, because POMO is an incremental solution generation method that relies on repeatedly re-encode the entire instance across multiple sampling rounds, making it more computation consuming. Compared to EGS, our method achieves similar or even better solution quality. EGS leverages a heuristic algorithm to solve subgraphs induced by edge generation, which provides a strong upper bound in optimization. However, EGS cannot guarantee superior performance, because the search space is significantly large, making it prone to local optima. Furthermore, EGS exhibits longer runtime due to its iterative edge resampling and repeated graph encoding. In summary, our method demonstrates consistent advantages over existing approaches in both solution quality and runtime efficiency across various DAG scheduling tasks.

# 7 Conclusion

In this paper, we present a novel one-shot solution generation method for DAG scheduling based on the policy gradient algorithm. We propose to identify the comparable antichain for each node during the topological sorting process, eliminating redundant comparisons and sampling computations, enabling more accurate gradient estimation. Furthermore, we design a dense reward that significantly improves training efficiency by mitigating the reward sparsity problem commonly observed in learning-based DAG scheduling. Experimental results across diverse scenarios demonstrate that our method achieves superior scheduling quality compared to existing DAG scheduling approaches. This suggest that our method have potential for further extension and optimization in large-scale or dynamic environments, like scientific workflow management. As future work, we will expand this foundational study to more specific applications by considering domain knowledge and optimization objectives.

**Limitations.** Although the experiment results present effectiveness of our method across various DAG scheduling tasks, the proposed comparable antichain identification approach is primarily designed to handle precedence constraints. Since our study is a continuation of prior foundational researches, we primarily focus on optimizing makespan objective. Extending our study to specific application by considering domain-specific objectives would be a promising direction. Moreover, for the scheduling problems involving domain-specific constraints, such as the deadline constraints in real-time systems, adaptations would be necessary. Additionally, our approach remains within the list scheduling paradigm. Exploring sequence-agnostic alternatives, such as one-shot edge generation for node prioritization, may be a promising direction.

## Acknowledgments and Disclosure of Funding

This research is supported in part by the project of National Natural Science Foundation of China (No. 52478030), the project of Shanghai Municipal Commission of Economy and Informatization (No. kz0023020251762), and the project of the Fundamental Research Funds for the Central Universities (No. 22120250610).

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

# A Illustration of our method

Our proposed framework is illustrated in Figure 3a, comprising two key components: comparable antichain identification (top-right) and dense reward estimation (bottom-left). The comparable antichain identification module constructs a complete global solution in the form of a node execution order, and estimates the sampling probability for each node. This solution is then simulated and evaluated to obtain nodewise cost values. By subtracting these with the cost values from a heuristic baseline solution and conducting normalization operation, we obtain nodewise reward signals. Figure 3b further provides a detailed illustration of the comparable antichain identification and sampling probability estimation process. At each step of the priority topological sorting, the set of zero in-degree nodes in the current subgraph is identified as a comparable antichain. Within this antichain, the node with the highest perturbed logits is selected for scheduling. The original logits are then passed through a softmax operation to compute sampling probabilities. This process continues iteratively to construct the full execution order along with the sampling probabilities for all nodes.

Note that the entire procedure requires only a single forward pass of the reinforcement learning network on the DAG scheduling problem instance, from which all logits are obtained in one shot.

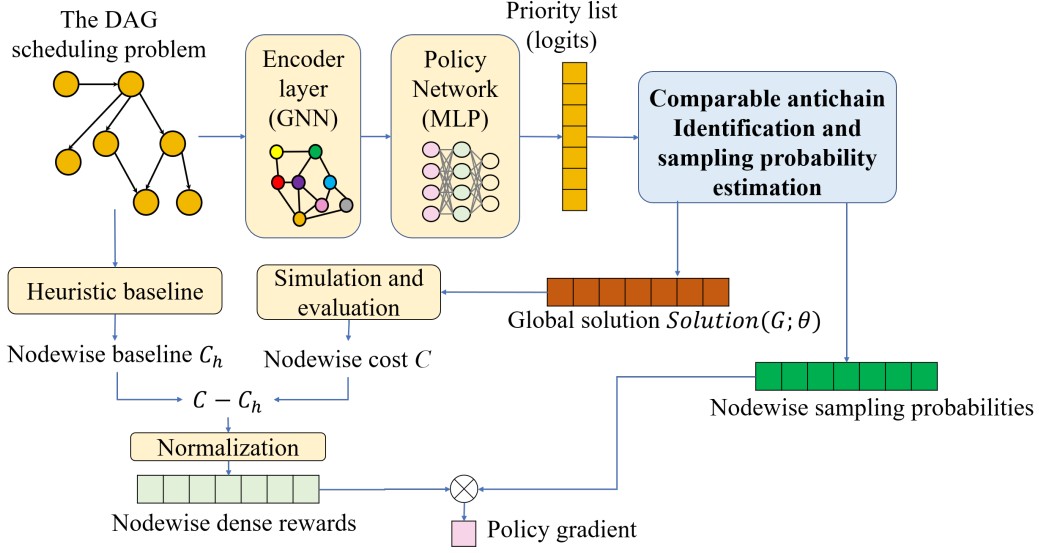

(a) Illustration of the overall framework.

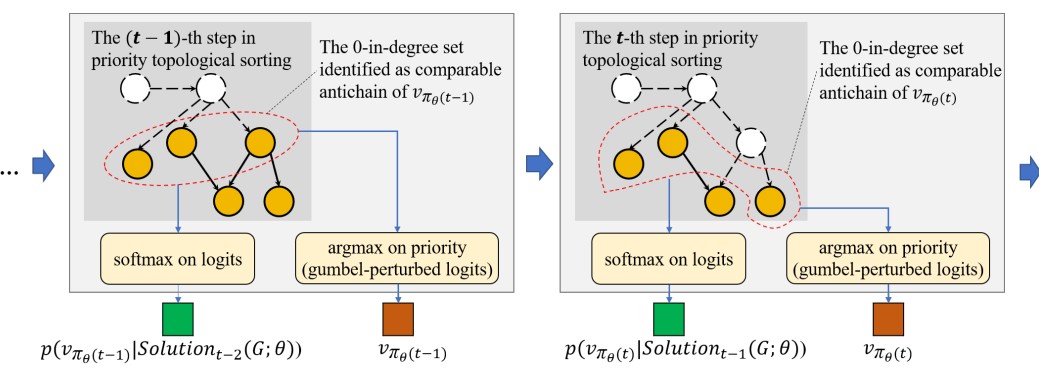

(b) Illustration of comparable antichain identification and sampling probability estimation.

Figure 3: Illustration of the proposed method.

# B   More information about the benchmarks

## B.1   Pegasus

Pegasus [Deelman et al., 2015] [2] provides an open-source workflow trace data generated from various scientific computing applications. We adopt it as the benchmark for DAG workflow scheduling under the heterogeneous multiprocessor setting. Under such setting, the attributes of task node $v_i$ include the computational workload $c_i$ and the output data size $b_i$. For each processor $m$, the key attribute is its computational capacity $f_m$. Assuming that task node $v_i$ is assigned to processor $m$, the processing time $d_i$ (as described in section 2) can be calculated as Equation (11).

$$d_i = \frac{c_i}{f_m} \tag{11}$$

Besides, Pegasus benchmark requires to consider the transmission time $z_i$ between processors: a task node cannot begin execution until all of its predecessor nodes have completed both their computation, and the transmission of their output data to the processor on which it is scheduled. Specifically, if a task node $v_i$ and its successor $v_j$ are assigned to different processors, then a transmission time is related to the output data size of $v_i$ and the bandwidth of computing environment. If both $v_i$ and $v_j$ are assigned to the same processor, the transmission time is considered negligible. This definition of $z_i$ is formalized in Equation (12).

$$z_i = \begin{cases} \frac{b_i}{\text{bandwidth}}, & \text{if } v_i \text{ and } v_j \text{ are on different processors} \\ 0, & \text{if } v_i \text{ and } v_j \text{ are on the same processor} \end{cases} \tag{12}$$

Our proposed method outputs only the execution order of the nodes, while the assignment of each node to a processor is determined using the Earliest Finish Time (EFT)-greedy rule. Specifically, for a given node to be scheduled, which is determined by the RL network, the EFT-greedy rule dispatches it to the processor that results in the earliest possible finish time. We adopt HEFT [Topcuoglu et al., 2002] as the advantage baseline for scheduling workflows in the Pegasus benchmark, as a part of the dense reward estimation. HEFT is a classic list-based heuristic algorithm for DAG scheduling on heterogeneous processors. It computes the priority (rank-up) of each node based on the average finish time of its successor nodes, and then assigns each node to a processor using the EFT-greedy rule.

## B.2   TPC-H

We adopt TPC-H as the benchmark for DAG workflow scheduling under the homogeneous single-processor setting. There is no need to dispatch tasks to specific processors in such setting. Each task node $v_i$ has a fixed processing time $d_i$ and computation resource requirement $q_i$. The total resource consumption of concurrently running tasks must not exceed the system's maximum resource capacity. We use the open source code [3] implemented by Wang et al. [2021] to generate TPC-H instances. Shortest Time First (STF) heuristic is adopted as the advantage baseline for TPC-H in our research. At each decision point, the STF rule selects the task with the shortest processing time.

## B.3   JSSP

The JSSP is a typical COP where each job consists of a sequence of operations. The operations within a job must follow a predefined order, forming precedence constraints, which can be modeled into a DAG. Different from the above-mentioned problem setting, each operation in JSSP must be processed on a specific machine for a specified duration. The scheduler needs to determine the execution order of operations that require the same machine but belong to different jobs. Consequently, JSSP is a special case of DAG scheduling problem. JSSP is often used as a standardized benchmark scenario for evaluating the performance of DAG scheduling methods, serving to validate their effectiveness on structurally constrained tasks. We use the code by Zhang et al. [2020] [4] to generate JSSP instances.

Similar with other DAG scheduling tasks, list scheduling can be applied to JSSP. By assigning each operation a priority value (or priority rank), the execution order of operations that require the same

---

[2] https://pegasus.isi.edu/workflow_gallery/
[3] https://github.com/Thinklab-SJTU/PPO-BiHyb/tree/main/dag_data/tpch
[4] https://github.com/zcaicaros/L2D/blob/main/DataGen/

machine can be determined. That is, if multiple operations compete for the same machine, the one with higher priority is selected to be executed first.

Shortest Processing Time (SPT) heuristic is adopted as the advantage baseline for JSSP. At each decision point, the SPT rule selects the operation with the shortest processing duration, and it would start on its designated machine at the earliest feasible time.

## C   Implementation details

### C.1   Input of network

A DAG scheduling problem instance consists of a DAG $G = (V, E)$ and a set of node attributes. These attributes typically include processing time $d_i$, resource or machine requirements, in-degree and out-degree, among others. Specifically, for DAG workflow scheduling under heterogeneous multiprocessors environments(as in the Pegasus benchmark), additional attributes include the required computation workload $c_i$, output data size $b_i$ of each node, and the computational capacity of each processor. For each task node $v_i$, we pack and normalize all these attributes into a vector $x_i$ as the raw feature of $v_i$, which is then fed into the GNN encoder.

### C.2   Neural network and hyper parameter settings

We used a Graphormer[Ying et al., 2021] with 4 layers and 4 attention heads. It outputs 64-dimension node embeddings for each task nodes. The policy network is a 64*64*1 MLP with ReLU activation function. We train each benchmark for at most 1000 epochs, and the batch size is 16. The logits regularization rate is 0.001. The Adam optimizer is employed with learning rate $5 \times 10^{-4}$.

### C.3   Perturbed logits

We follow Jeon et al. [2023] to introduce Gumbel trick [Kool et al., 2019] when sampling the node execution order. Instead of treating the original logits output by the policy network as the priority values of the task nodes, the argmax sampling operation is performed over the comparable antichain $V_t^{in=0}$ on perturbed logits (formulated in Equation (13)), which follows on Gumbel distribution (formulated in Equation (14)). Therefore, the sampling becomes more aligned with the underlying probability distribution formulated in Equation (6).

$$\text{perturbed\_logits}(v_i) = \text{logits}(v_i) + Z(v_i) \tag{13}$$

$$Z(v) = -\log(-\log p), p \sim \text{Uniform}(0, 1) \tag{14}$$

### C.4   Simulation and evaluation environment

To evaluate the generated scheduling solutions and obtain both the overall optimization objective and node-level dense rewards, we implemented a DAG workflow simulation environment based on the open-source SimPy [5] platform based on Python language. This simulator is further wrapped into an OpenAI Gym[6] environment to integrate with reinforcement learning frameworks. It is capable to simulate all three aforementioned benchmarks, and can be extended to support other scheduling scenarios if necessary.

## D   Dense reward signal calculation procedure

Our dense reward is constructed by simulating the entire one-shot generated schedule, and evaluating the global cost (e.g., makespan). Then, we compute a node-level reward for each scheduled node, based on its distance to the final cost. Specifically:

---

[5] https://simpy.readthedocs.io/en/
[6] https://github.com/openai/gym

1. Given a DAG scheduling problem $G$, we conduct the heuristic advantage baseline algorithm on $G$, obtaining each task node's individual baseline cost $C_h(v_i)$.

2. For a sampled solution $Solution(G; \theta)$ (i.e., an execution order), we simulate it using a SimPy-based simulator. and obtain the overall makespan $C(Solution(G; \theta))$ in practice. For each task node $v_{\pi(t)}$, we obtain its individual cost $C(S_t(G; \theta))$ (e.g., its finish time) through simulation.

3. We obtain the return-like dense reward signal $R_t$ of each task node $v_{\pi(t)}$ by comparing the global objective $C(Solution(G; \theta))$ with individual cost $C(S_t(G; \theta))$, according to Equation (8).

4. To derive the advantage-like feedback $A_t$, we further substract each baseline from $R_t$, according to Equation (9).

## E   Baseline algorithms

**Jeon et al. [2023].**   Although the original authors did not fully release their source code, the idea described in their paper is sufficiently clear and straightforward to reproduce. So we implemented their approach accordingly.

**POMO-DAG.**   We build POMO-DAG upon the POMO [7] proposed by Kwon et al. [2020], adapting its problem instance encoder to a Graphormer-based GNN[Ying et al., 2021] so that it can process DAG scheduling problems. The model generates a ranked list of nodes as the scheduling solution.

**EGS.**   For EGS [Sun et al., 2024], the basic framework of the original code is publicly available[8]. We retained the original structure and implemented the missing policy network and training procedure that were not released.

## F   Additional results on Pegasus benchmark

Due to page limits, additional experimental results on the Pegasus benchmark are presented here in the appendix, specifically for the LIGO and GENOME datasets (see Table 4 and Table 5, respectively). Similar to the results on SIPHT presented in section 6, we selected 5 different problem instances of each dataset (100, 200, 300, 400, and 1000 task nodes). The evaluation metrics include the optimization objective (makespan, or MS in short), the gap relative to the heuristic baseline (HEFT), and the run time.

In the LIGO-1000 case, neither our method nor the neural baselines outperformed the heuristic algorithm. We attribute this to the fact that the result (2373.5) obtained by the heuristic method is already close to the lower bound of the optimization objective, leaving little room for improvement.

## G   The impact of heuristic advantage baseline selection

Considering that the selection of heuristic algorithm for advantage baseline might influence the effectiveness of dense reward signal, we conducted experiments comparing multiple heuristic advantage baselines, including the previously used HEFT, and Critical Path on a Processor (CPOP), Shortest Finish Time (SFT). The results of the gap related to HEFT in percentage is shown in the following Table 6, as a complement to Table 1. The results show that our method with different advantage baseline still outperforms existing methods, and the performance of our method with different heuristic advantage baseline is comparable. We believe this is because the heuristic algorithm provide constant estimates for each DAG scheduling problem instance, ensuring the advantage estimation is unbiased. Additionally, these heuristics are near-optimal in many cases, leading to similar schedules. As a result, the variance reduction benefit is preserved, while introducing little bias.

---

[7]https://github.com/yd-kwon/POMO
[8]https://github.com/binqi-sun/egs

Table 4: Experimental results on LIGO dataset in Pegasus.

| Method | LIGO-100 | | | LIGO-200 | | | LIGO-300 | | | LIGO-400 | | | LIGO-1000 | | |
|---|---|---|---|---|---|---|---|---|---|---|---|---|---|---|---|
| | MS | Gap/% | Time/s | MS | Gap/% | Time/s | MS | Gap/% | Time/s | MS | Gap/% | Time/s | MS | Gap/% | Time/s |
| HEFT (baseline) | 217.5 | - | 0.005 | 462.9 | - | 0.014 | 670.3 | - | 0.026 | 959.3 | - | 0.043 | **2373.5** | - | 0.221 |
| Jeon et al. [2023] | 217.0 | -0.23 | 0.08 | 465.1 | 0.47 | 0.16 | 670.8 | 0.07 | 0.26 | 962.8 | 0.36 | 0.43 | 2376.7 | 0.13 | 2.39 |
| POMO-DAG | 216.7 | -0.37 | 11.43 | 473.8 | 2.35 | 17.50 | 675.3 | 0.75 | 23.04 | 969.1 | 1.02 | 28.48 | 2382.0 | 0.36 | 50.56 |
| EGS | 214.2 | -1.52 | 0.98 | 462.5 | -0.09 | 3.07 | 669.9 | -0.06 | 5.34 | 956.9 | -0.25 | 8.75 | **2373.5** | 0 | 55.14 |
| Ours | **214.0** | **-1.61** | 0.62 | **462.1** | **-0.17** | 0.99 | **668.8** | **-0.22** | 1.20 | **956.6** | **-0.28** | 1.26 | **2373.5** | 0 | 2.59 |
| Ours w/o DR | 214.0 | -1.56 | 0.36 | 464.5 | 0.34 | 0.70 | 670.0 | -0.04 | 1.45 | 957.9 | -0.14 | 1.93 | 2375.5 | 0 | 2.96 |
| Ours w/o CAI | 215.9 | -0.94 | 0.10 | 462.7 | -0.04 | 0.25 | 672.1 | 0.13 | 0.32 | 960.4 | 0.11 | 0.48 | 2377.6 | 0.17 | 2.12 |

Table 5: Experimental results on GENOME dataset in Pegasus.

| Method | GENOME-100 | | | GENOME-200 | | | GENOME-300 | | | GENOME-400 | | | GENOME-1000 | | |
|---|---|---|---|---|---|---|---|---|---|---|---|---|---|---|---|
| | MS | Gap/% | Time/s | MS | Gap/% | Time/s | MS | Gap/% | Time/s | MS | Gap/% | Time/s | MS | Gap/% | Time/s |
| HEFT (baseline) | 2553.1 | - | 0.005 | 2373.4 | - | 0.013 | 4751.9 | - | 0.028 | 3453.2 | - | 0.043 | 15016.8 | - | 0.239 |
| Jeon et al. [2023] | 2511.4 | -1.63 | 0.08 | 2369.9 | -0.15 | 0.15 | 4755.3 | 0.07 | 0.27 | 3483.2 | 0.87 | 0.46 | 15001.9 | -0.10 | 2.58 |
| POMO-DAG | 2472.0 | -3.18 | 11.32 | 2367.2 | -0.26 | 17.42 | 4783.2 | 0.66 | 22.83 | 3527.6 | 2.16 | 29.04 | 15005.7 | -0.07 | 50.55 |
| EGS | 2475.6 | -3.04 | 0.94 | 2356.2 | -0.72 | 4.05 | 4730.0 | -0.46 | 6.43 | 3453.0 | -0.01 | 9.08 | 14970.4 | -0.31 | 71.90 |
| Ours | **2468.2** | **-3.32** | 0.61 | **2350.5** | **-0.96** | 1.00 | **4728.4** | **-0.49** | 1.20 | **3453.0** | **-0.01** | 1.29 | **14955.8** | **-0.41** | 2.79 |
| Ours w/o DR | 2507.6 | -1.78 | 0.36 | **2350.5** | **-0.96** | 0.70 | 4744.1 | -0.16 | 1.45 | 3453.2 | 0 | 1.93 | 14978.3 | -0.26 | 2.96 |
| Ours w/o CAI | 2495.6 | -2.25 | 0.11 | 2358.6 | -0.62 | 0.22 | 4734.9 | -0.36 | 0.34 | 3453.2 | 0 | 0.49 | 14984.2 | -0.22 | 2.37 |

Table 6: Experimental results on relative gap about the selection of heuristic advantage baseline.

| Heuristic Method | SIPHT-100 | SIPHT-200 | SIPHT-300 | SIPHT-400 |
|---|---|---|---|---|
| **ours + HEFT** | -13.3 | -5.42 | -0.25 | -0.62 |
| **ours + CPOP** | -12.2 | -5.42 | -0.20 | -0.62 |
| **ours + SFT** | -12.8 | -4.83 | -0.22 | -0.62 |

# H  Convergence guarantee analysis

Since the optimization of our approach is based on policy gradient, it inherits the theoretical convergence properties of the classical policy gradient framework. Although the convergence performance in actual usage might be not as perfectly good as that of the classic policy gradient under ideal conditions, several designs in our method could enhance the stability and convergence behavior in practice:

- 1. Nodewise dense rewards signal provides a closer approximation of value/advantage estimation at each decision step.
- 2. The CAI module improves the quality of nodewise decision probability estimation, obtaining more accurate gradients and hence improved convergence behavior.
- 3. For a given DAG scheduling instance, the results generated by the heuristic advantage baseline algorithm is fixed across training epochs. This ensures that the advantage estimation is unbiased.

While these designs do not provide formal theoretical bounds, they provide empirical support for training stability and convergence, as reflected in the performance across diverse testbenches.

# I  Interpretability analysis

Darvariu et al. [2024] pointed out that one-shot RL methods for combinatorial optimization problems tend to be less interpretable than incremental solution construction approaches, as they do not strictly follow the Markov decision process and do not satisfy the Bellman equation in value evaluation. However, by introducing the dense reward signal, our method improves the interpretability of one-shot approaches to some extent. Specifically, the nodewise decisions, sampling probabilities and dense reward signals can be regarded as an entire MDP sampling trajectory in RL (like a trajectory by

Monte Carlo sampling). This makes the interpretability of one-shot learning closer to MDP-based incremental approaches than those one-shot methods with sparse rewards.

