# OpenReview forum: "Reinforcement learning for one-shot DAG scheduling with comparability identification and dense reward"
_NeurIPS.cc/2025/Conference — NeurIPS 2025 poster_

### Official Review · Reviewer_WSCD · 2025-06-27

**Clarity:** 3
**Significance:** 3
**Originality:** 3
**Rating:** 4
**Confidence:** 4

**Summary:**

This paper shows a one-shot reinforcement learning (RL) method for scheduling tasks represented as a Directed Acyclic Graph (DAG). The authors address two key issues in existing one-shot approaches: redundant priority comparisons among nodes on the same precedence path, leading to biased probability estimates and unstable training, and sparse global rewards that offer little guidance for individual node-level decisions. To overcome these challenges, they propose a CAI mechanism, which extracts only nodes whose relative ordering affects the final schedule, and a dense reward signal that decomposes global makespan cost into node-level contributions. Empirical evaluations across three benchmarks—Pegasus scientific workflows, TPC-H query DAGs, and job-shop scheduling instances—demonstrate that the proposed method outperforms state-of-the-art one-shot RL, edge-generation scheduling, and incremental baselines in training efficiency.

**Questions:**

A few questions that could improve the article:

Q: Have you tested the method on objectives beyond makespan? example: cumulative flow time
Comparable improvements with alternative cost functions would support broader applicability and show that both CAI and DR are agnostic to objective design.

Q: Can the method handle settings where the DAG is revealed incrementally or changes over time?
It would be interesting to know extensions in dynamic environments and outlining path for adaptation.

**Ethical Concerns:**

["NO or VERY MINOR ethics concerns only"]

**Final Justification:**

I would keep the score, since there aren't additional prelim results to address the weaknesses

**Limitations:**

yes

**Quality:**

3

**Strengths And Weaknesses:**

The strengths are:
-The work demonstrates gains across heterogeneous benchmarks, which highlights robustness
-The method applies node-level dense rewards based on heuristic advantage baselines in one-shot RL, that is an adaptation of reward-shaping concepts.

The weaknesses are:
- The paper does not explore other objective metrics or multi-objective scenarios, which narrows its applicability.

---

> ### Author Rebuttal · Authors · 2025-07-31
>
> We thank the viewer for careful evaluation of our manuscript and the valuable feedback. In this rebuttal, we respond to the concerned issues and clarify several aspects of our work.
>
> #### Weakness 1:
> >The paper does not explore other objective metrics or multi-objective scenarios, which narrows its applicability.
>
> In this work, our main goal is to propose a foundational scheduling framework, rather than a specialized model for a particular domain. Our approach can be viewed as an extension of prior foundational DAG scheduling studies (e.g., Jeon et al. [2023], Sun et al.[2024] referenced in the manuscript) , where makespan is adopted for evaluating DAG scheduling algorithms. Therefore, adopting makespan as the metric enables us to demonstrate direct comparison with prior works.
>
> We regard extending the proposed one-shot scheduling method to application-specific objectives as a promising future direction, by considering domain knowledge (such as the highly real-time system model, edge computing, and quality-of-service objectives). We will add the corresponding description in the Conclusion section.
>
> #### Question 1:
> >Have you tested the method on objectives beyond makespan? example: cumulative flow time. Comparable improvements with alternative cost functions would support broader applicability and show that both CAI and DR are agnostic to objective design.
>
> As explained in the response to Weakness 1, we only tested performance on makespan. We appreciate the viewer's thoughtful suggestion to consider other objective metrics or multi-objective scenarios to support broader applicability of our proposed method. We agree that this would be a promising direction.
>
> #### Question 2:
> >Can the method handle settings where the DAG is revealed incrementally or changes over time? It would be interesting to know extensions in dynamic environments and outlining path for adaptation.
>
> While our proposed one-shot policy is evaluated in relatively static settings, it can be applied to dynamic environments. For example, when new task nodes are revealed, or a new DAG job instance is submitted, re-scheduling can be performed by re-running the network. The optimization objective and dense reward signal calculation should be adjusted accordingly. In cases where DAG job structures and arrival times are known in advance (i.e., containing earliest-start-time constraints), we can perform pre-processing by adding pseudo predecessor nodes with fixed execution times.
>
> Considering that real-world dynamic scheduling scenarios often involve domain-specific constraints, it might require more dedicated modeling. Therefore, we think this is a highly promising future work, and we will accordingly add the discussion in the Conclusion section.

---

> ### Comment · Reviewer_WSCD · 2025-08-02
> **Thanks for clarification**
>
> Thank you for clarification and explanations of the questions.
> Since there aren't additional prelim results to address the weaknesses, I would keep the score.

---

### Official Review · Reviewer_GVMe · 2025-07-01

**Clarity:** 3
**Significance:** 3
**Originality:** 2
**Rating:** 4
**Confidence:** 3

**Summary:**

This paper proposes a method using comparable antichain identification to reduce redundant comparisons and a dense reward signal for node-level optimization, reporting improved scheduling objective results compared to other learning-based approaches.

**Questions:**

Please see the weaknesses.

**Ethical Concerns:**

["NO or VERY MINOR ethics concerns only"]

**Final Justification:**

The rebuttal resolves my major concerns.

**Limitations:**

yes

**Quality:**

2

**Strengths And Weaknesses:**

Strengths

1.The paper presents a structurally guided one-shot DAG scheduling framework that integrates node-level decision mechanisms into a policy gradient setup.

2.It provides a clear analysis of the limitations of prior one-shot approaches, particularly their reliance on redundant node comparisons, and proposes targeted improvements to address them.

Weaknesses

1.The ablation results in Table 2 show that removing either CAI or the dense reward sometimes leads to better performance than using both modules together. This suggests potential negative interactions or lack of synergy between the components, raising questions about the robustness of the combined design, especially across different DAG structures such as shallow or disjoint graphs.

2.The motivation for the CAI module is based on the assumption that redundant comparisons lead to inaccurate sampling probability. However, this claim is primarily supported by intuitive reasoning and illustrative examples (e.g., Fig. 2), lacking formal justification or empirical analysis. A more rigorous treatment of how CAI affects the sampling distribution or improves policy gradient estimation would enhance the technical depth and clarity of the contribution.

---

> ### Author Rebuttal · Authors · 2025-07-31
>
> We thank for the viewer for your careful reading and valuable suggestion of our manuscript. In this rebuttal, we would like to respond to the concerns raised by the reviewer.
>
> #### Weakness & Question 1:
> > The ablation results in Table 2 show that removing either CAI or the dense reward sometimes leads to better performance than using both modules together. This suggests potential negative interactions or lack of synergy between the components, raising questions about the robustness of the combined design, especially across different DAG structures such as shallow or disjoint graphs.
>
> In these specific cases, the performance degradation caused by introducing CAI is more. We attribute this to two main reasons:
>
> 1. The DAGs in a TPC-H scheduling problem instance exhibit imbalanced structures. Also, since these DAGs are not mutually interconnected, the entire scheduling instance is a highly sparse graph structure. Moreover, the number of DAGs in the TPC-H benchmark is relatively limited. As a result, GNN-based network may struggle to learn to capture the features of each node across all DAGs within the scheduling instance. In addition, since CAI masks out incomparable nodes at each step, it may further hinder the network from learning accurate global priority value (logits) for each node.
>
> 2. Our proposed CAI intends to remove the redundant priority comparison among DAG nodes that has dependency relationship, thus obtaining more accurate sampling probability. So, CAI is particularly useful in DAGs with more complex dependency structures, in which multiple nodes often lie along a longer path. For simpler DAG structures (e.g., TPC-H benchmark), they typically consist of many small-size DAGs with much shorter paths. In this situation, the number of dependent nodes located on the same path is small, and the existing dependencies are weak. The redundant comparisons among dependent nodes are significantly reduced, so the further improvement from applying CAI is limited.
>
> In future work, we plan to investigate strategies for better handling such cases, including more effective feature extraction or training schemes tailored for such multi-DAG instances.
>
> #### Weakness & Question 2:
> >The motivation for the CAI module is based on the assumption that redundant comparisons lead to inaccurate sampling probability. However, this claim is primarily supported by intuitive reasoning and illustrative examples (e.g., Fig. 2), lacking formal justification or empirical analysis. A more rigorous treatment of how CAI affects the sampling distribution or improves policy gradient estimation would enhance the technical depth and clarity of the contribution.
>
> We appreciate the reviewer’s request for a more rigorous explanation of how CAI improves sampling probability estimation and policy gradient updates. We provide a clarification below.
>
> Existing methods typically sample a ranking list over all nodes (i.e., a total order) and conduct a topological sort to get the the actual node execution order as the solution from the rank list. However, multiple distinct ranking lists $\{Rank_1, Rank_2, \dots\}$ can lead to a same valid DAG schedule $Solution(G)$, and the total probability of sampling that solution should be:
> $$
> P(Solution(G)) = P(Rank_1 \cup Rank_2 \cup …)
> $$
>
> So, if only a single rank list’s probability $P(Rank_j)$ is used to approximate $P(Solution(G))$, this introduces underestimation. This increases entropy and the variance of the policy gradient.
>
> We believe the reason causing the above issue lies in the sampling process. Existing methods use Equation (2) in the manuscript in each ranking step, which involves choosing each node among all unranked nodes. This regards the nodes that cannot possibly be scheduled at that step (i.e., nodes that are on the same dependency path with the currently selected node) as candidate nodes. These comparisons are thus redundant or invalid, as they do not affect the resulting topological order. Treating them as candidate actions introduces irrelevant logits into the softmax distribution, distorting the sampling probability.
>
> Our CAI module first restricts comparisons to independent nodes. Then, among these mutually independent nodes, it further selects the subset that actually influences the final scheduling order — the comparable antichain. This masks those invalid nodes, making the actual sampling distribution closer to the true $P(Solution(G))$.  As a result, it reduces entropy and variance of policy gradient.
>
> We will revise Section 3 and 4 to include the above explanation. We thank the reviewer's suggestion for making this point clearer and more rigorous.

---

> > ### Comment · Reviewer_GVMe · 2025-08-06
> >
> > Thank you for your detailed rebuttal addressing the concerns raised. The rebuttal resolves my major concerns.

---

### Official Review · Reviewer_ZH3U · 2025-07-02

**Clarity:** 2
**Significance:** 3
**Originality:** 2
**Rating:** 4
**Confidence:** 3

**Summary:**

This paper addresses critical limitations in existing reinforcement learning (RL) approaches for one-shot Directed Acyclic Graph (DAG) scheduling. Current RL-based methods suffer from two fundamental issues: redundant node priority comparisons that lead to biased sampling probability estimation, and sparse reward signals that hinder effective policy optimization. To overcome these challenges, we introduce a novel framework featuring a comparable antichain identification mechanism and a dense reward design. The comparable antichain identification dynamically determines which node pairs actually influence scheduling outcomes during topological sorting, eliminating unnecessary priority comparisons and enabling accurate probability estimation for gradient computation. This is complemented by a dense reward signal that provides node-level optimization guidance by comparing each node's completion time against heuristic baselines (e.g., HEFT for Pegasus, SFT for TPC-H), effectively mitigating the sparse reward problem.

**Questions:**

1.  How does the comparable antichain identification mechanism specifically enhance the accuracy of nodewise sampling probabilities compared to existing methods?
2. Can the authors provide more details on how the dense reward signal is calculated and how it improves training efficiency in practice?

**Ethical Concerns:**

["NO or VERY MINOR ethics concerns only"]

**Final Justification:**

The problem is interesting, and I think some parts can be further clarified, like the importance of using onepass manner to determine a DAG. I still think high computational load is not so convincing since DAG may be data dependent and adjustable each round. Overall I would like to recommend borderline acc.

**Limitations:**

While the proposed method demonstrates effectiveness on the tested benchmarks, the paper lacks a detailed discussion on its scalability for very large-scale DAG scheduling problems. As the scale of DAGs increases, the computational overhead of the comparable antichain identification mechanism may become significant. The authors could further explore strategies to enhance the method's scalability for large-scale scenarios.

**Paper Formatting Concerns:**

No formatting issues were found after checking against NeurIPS guidelines.

**Quality:**

3

**Strengths And Weaknesses:**

Strengths:
1.The paper proposes a novel one-shot DAG scheduling method based on reinforcement learning, introducing a comparable antichain identification mechanism and a dense reward signal. This effectively addresses the issues of redundant node priority comparisons and sparse rewards in existing RL-based one-shot scheduling approaches. The comparable antichain identification mechanism eliminates unnecessary pairwise priority comparisons, improving the accuracy of sampling probability estimation and training stability. The dense reward signal provides node-level optimization guidance, alleviating the sparse reward challenge and enhancing training efficiency.
2. The paper provides a detailed analysis of the limitations of existing methods, including redundant priority comparisons and biased sampling probability estimation, as well as the sparse reward challenge. It demonstrates through theoretical derivations and lemma proofs that the comparable antichain identification method can accurately reflect the nodewise sampling probability corresponding to the sampled node execution sequence. This offers a solid theoretical foundation for the proposed method.
3. The experimental section is well-designed, with evaluations conducted on three different DAG scheduling benchmarks (Pegasus, TPC-H, and JSSP). The results demonstrate that the proposed method achieves superior scheduling quality compared to existing learning-based DAG scheduling approaches.Ablation studies further validate the effectiveness of the comparable antichain identification mechanism and dense reward signal, highlighting their respective contributions to the overall performance.
4. The paper makes several important contributions to the field of DAG scheduling. It not only proposes an innovative RL-based one-shot scheduling method but also provides new insights and solutions for addressing the challenges of redundant priority comparisons and sparse rewards in RL-based scheduling. This advances the development of reinforcement learning in DAG scheduling applications.

Weaknesses:
1. Although the dense reward signal improves training efficiency, the paper does not provide a thorough theoretical analysis of its design principles and theoretical properties. For instance, the impact of the advantage baseline selection on the dense reward signal's performance and its adaptability across different scheduling scenarios could be explored in greater depth.
2. The paper compares the proposed method with several existing approaches but lacks a comprehensive comparison with the latest state-of-the-art methods in different application domains. This makes it difficult to fully assess the proposed method's competitiveness and generalizability. Including comparisons with more advanced methods would better highlight the method's advantages and limitations.

---

> ### Author Rebuttal · Authors · 2025-07-31
>
> We thank the viewer for thoughtful and careful evaluation of our manuscript and the valuable feedback. In this rebuttal, we respond to the concerned issues and clarify several aspects of our work.
>
> #### Weakness 1:
> > Although the dense reward signal improves training efficiency, the paper does not provide a thorough theoretical analysis of its design principles and theoretical properties. For instance, the impact of the advantage baseline selection on the dense reward signal's performance and its adaptability across different scheduling scenarios could be explored in greater depth.
>
> We appreciate the reviewer’s insightful comment regarding the design and theoretical properties of the dense reward signal.
>
> The core motivation for using dense rewards is to improve training efficiency and convergence stability in one-shot policy gradient learning, where the traditional sparse final reward often results in poor credit assignment. To address this, we design a dense reward signal by decomposing the total objective into node-wise costs. It is computed through simulation. For each scheduled node, we define its dense reward signal based on its distance to the final objective, thus approximating the node's value estimation when it is executed. Such design also makes one-shot DAG scheduling more interpretable. Specifically, the nodewise decisions, sampling probabilities and dense reward signals can be regarded as an entire MDP sampling trajectory in RL (like a trajectory by Monte Carlo sampling). This makes the interpretability of one-shot learning closer to MDP-based incremental approaches than those one-shot methods with sparse rewards.
>
> As for the impact of the advantage baseline selection, we conducted experiments comparing multiple heuristic advantage baselines, including the previously used HEFT, and Critical Path on a Processor (CPOP), Shortest Finish Time (SFT). The results of the gap related to HEFT in percentage is shown in the following table, as a complement to Table 1. The results show that our method with different advantage baseline still outperforms existing methods, and the performance of our method with different heuristic advantage baseline is comparable.
>
> |Method |SIPHT100|SIPHT200|SIPHT300|SIPHT400|
> |--------| -----| -------|--------|------|
> |ours+HEFT|-13.3 |-5.42  |  -0.25 |  -0.62|
> |ours+CPOP|-12.2 |-5.42  |  -0.20 | -0.62|
> |ours+SFT |-12.8 |-4.83  |  -0.22 | -0.62|
>
> We believe this is because the heuristic algorithm provide constant estimates for each DAG scheduling problem instance, ensuring the advantage estimation is unbiased. Additionally, these heuristics are near-optimal in many cases, leading to similar schedules. As a result, the variance reduction benefit is preserved, while introducing little bias.
>
> #### Weakness 2:
> > The paper compares the proposed method with several existing approaches but lacks a comprehensive comparison with the latest state-of-the-art methods in different application domains. This makes it difficult to fully assess the proposed method's competitiveness and generalizability. Including comparisons with more advanced methods would better highlight the method's advantages and limitations.
>
> In this work, our main goal is to propose a foundational scheduling framework, rather than a specialized model for a particular domain. Our approach can be viewed as an extension of prior foundational DAG scheduling studies (e.g., Jeon et al. [2023], Sun et al.[2024] referenced in the manuscript). We focus on comparisons with these recent and representative foundational studies.
>
> For future work, we regard adapting the proposed one-shot scheduling method to specific domain as a promising direction. The proposed CAI mechanism and dense reward signal can be extended to real-world systems by considering domain knowledge (such as the highly real-time system model, edge computing, and quality-of-service objectives) to better suit in specific areas. We will accordingly enrich the "Conclusion" section to discuss these domain adaptations and other future extensions.
>
> #### Question 1:
> > How does the comparable antichain identification mechanism specifically enhance the accuracy of nodewise sampling probabilities compared to existing methods?
>
> Existing methods typically sample a ranking list over all nodes (i.e., a total order) and conduct a topological sort to get the the actual node execution order as the solution from the rank list. However, multiple distinct ranking lists $\{Rank_1, Rank_2, \dots\}$ can lead to a same valid DAG schedule $Solution(G)$, and the total probability of sampling that solution should be:
> $$
> P(Solution(G)) = P(Rank_1 \cup Rank_2 \cup …)
> $$
>
> So, if only a single rank list’s probability $P(Rank_j)$ is used to approximate $P(Solution(G))$, this introduces underestimation.
>
> We believe the reason causing the above issue lies in the sampling process. Existing methods use Equation (2) in the manuscript in each ranking step, which involves choosing each node among all unranked nodes. This regards the nodes that cannot possibly be scheduled at that step (i.e., nodes that are on the same dependency path with the currently selected node) as candidate nodes. These comparisons are thus redundant or invalid, as they do not affect the resulting topological order. Treating them as candidate actions introduces irrelevant logits into the softmax distribution, distorting the sampling probability.
>
> Our CAI module first restricts comparisons to independent nodes. Then, among these mutually independent nodes, it further selects the subset that actually influences the final scheduling order — the comparable antichain. This masks those invalid nodes, making the actual sampling distribution closer to the true $P(Solution(G))$.
>
>
> #### Question 2:
> > Can the authors provide more details on how the dense reward signal is calculated and how it improves training efficiency in practice?
>
> Our dense reward is constructed by simulating the entire one-shot generated schedule, and evaluating the global cost (e.g., makespan). Then, we compute a node-level reward for each scheduled node, based on its distance to the final cost. Specifically:
>
> 1. Given a DAG scheduling problem $G$, we conduct the heuristic advatage baseline algorithm on $G$, obtaining each task node's individual baseline cost $C_h(v_i)$.
> 2. For a sampled solution $Solution(G;\theta)$ (i.e., an execution order), we simulate it using a SimPy-based simulator. and obtain the overall makespan $C(Solution(G;\theta))$ in practice. For each task node $v_{\pi(t)}$, we obtain its individual cost $C(S_t(G;\theta)) $ (e.g., its finish time) through simulation.
> 3. We obtain the return-like dense reward signal $R_t$ of each task node $v_{\pi(t)}$ by comparing the global objective $C(Solution(G;\theta))$ with individual cost $C(S_t(G;\theta))$:
> $$ R_t = C(Solution(G;\theta)) - C(S_t(G;\theta))$$
> 4. To derive the advantage-like feedback $A_t$, we further substract each baseline from $R_t$, obtaining:
> $$
> \begin{align*}
> A_t &= R_t - \textbf{baseline} \\
> &= (C(Solution(G;\theta)) - C(S_t(G;\theta))) - (C(Solution(G;\theta))-C_h(v_{\pi_\theta(t)})) \\
> &=C_h(v_{\pi_\theta(t)}) - C(S_t(G;\theta))
> \end{align*}
> $$
>
> By introducing heuristic baseline, we provide a meaningful reward signal for each action, rather than assigning the same reward to all nodes or only to the final outcome.
>
> As for how it improves training efficiency in practice, compared to sparse reward settings (where only the final cost is used), our dense reward reduces the variance of policy gradient estimates, because each node's policy update receives a targeted reward signal rather than sharing a single scalar. Also, it helps the model differentiate good and bad actions early in training.
>
> #### Limitation:
> >While the proposed method demonstrates effectiveness on the tested benchmarks, the paper lacks a detailed discussion on its scalability for very large-scale DAG scheduling problems. As the scale of DAGs increases, the computational overhead of the comparable antichain identification mechanism may become significant. The authors could further explore strategies to enhance the method's scalability for large-scale scenarios.
>
> CAI demonstrates better performance especially on those testbenches with more complex node dependency relationship. Our method also inherits the runtime advantages of one-shot scheduling compared to some other RL-based approaches, as demonstrated in the results. This suggest that our method have potential for further extension and optimization in large-scale environments, like scientific workflow management. We thank the reviewer's valuable suggestion which we consider a promising future extension direction.

---

> ### Comment · Reviewer_ZH3U · 2025-08-05
> **Responses to authors**
>
> Thank you for the responses. I still feel a bit puzzle about the commonness of the one shot learning in DAG discovery. Let us use the gradient based DAG learning as examples. We often fit the data until a model converges, and discover the best scored graph. In the agent case, it seems that one-shot is no need. I still can train an agent that specifically tie to one data distribution? Would that be the best way to apply RL for DAG learning?

---

> > ### Author Response · Authors · 2025-08-06
> >
> > Thank for your response. Our previous response about “one-shot learning” might cause ambiguity. We apologize if there is any confusion. We would like to further clarify as follows:
> >
> > In our paper, "one-shot" refers to the way to infer the solution of DAG scheduling. That is, all the scheduling decisions for a DAG scheduling problem are generated by a single forward propagation of the whole network consisting of the encoder sub-network and the policy sub-network. During our training process, the GNN encoder sub-network and logits policy sub-network are still fitted through multiple iterations.
> >
> > If the one-shot solution generation approach was not used, at each decision step, the model had to re-encode the status of the the whole DAG scheduling problem with the encoder sub-network, and then generate the current decision with the policy sub-network, so the scheduling decisions were made incrementally. This leads to high computational cost for both inference and training. So, from the perspective of computational cost for inference and training, the one-shot scheduling is necessary.

---

### Official Review · Reviewer_ebqq · 2025-07-03

**Clarity:** 2
**Significance:** 2
**Originality:** 2
**Rating:** 5
**Confidence:** 4

**Summary:**

The paper introduces a new one-shot method for solving the Directed Acyclic Graph (DAG) scheduling problem using reinforcement learning (RL). The paper identifies two primary weaknesses in existing one-shot scheduling approaches: the redundant priority comparisons among nodes, which leads to biased estimates of sampling probabilities, and sparse reward signals, which provide inefficient guidance for the learning agent.

The paper introduces Comparable Antichain Identification (CAI) mechanism, which eliminates unnecessary comparisons by only considering nodes whose relative priority actually impacts the final execution order.

The paper further presents a dense reward signal for RL with node-level feedback during training. using Policy Gradient for training the GNN encoder.

The model is benchmarked using three existing datasets (Pegasus, TPC-H, and JSSP) and compared against heuristic and learning based models. The experimental results show that the proposed method generally outperforms other learning-based approaches in terms of solution quality and is more computationally efficient than incremental methods.

**Questions:**

- What are the disadvantages of relying on Q-Learning for the generation of a dense reward? Could a simple feedback per scheduled task performacne be used to make the reward dense for each scheduling step?
- How sensitive is the method’s performance to the choice of the heuristic used for advantage baseline estimation?
- Are there theoretical guarantees (e.g., bounds) on performance improvements or convergence properties?

**Ethical Concerns:**

["NO or VERY MINOR ethics concerns only"]

**Limitations:**

- The method assumes access to a decent heuristic for baseline reward estimation.
- CAI's benefit is less pronounced on shallow or disjoint DAGs.
- Despite improved training efficiency, the GNN-based architecture and sorting overhead may still limit real-time or very large-scale applications.
- The method does not directly model dynamic environments or online scheduling, restricting its scope to static DAG instances.

**Paper Formatting Concerns:**

Limitations section discussing the model limitations is not clearly stated. Having a distinct paragraph or subsection would improve the quality of the paper.

**Quality:**

2

**Strengths And Weaknesses:**

Strengths
- Carefully analyzes limitations in prior work and supports its approach with formal definitions, lemmas, and justification.
- Empirical Results include Ablation study to show the impact of each component of the proposed method.
- The experiments are done on a broad range of problems.

Weaknesses
- The method underperforms or matches baselines on simpler DAG structures (e.g., TPC-H). While mentioned, this is not deeply analyzed.
- Like most one-shot methods, it sacrifices interpretability compared to MDP-based incremental approaches.
- Dense reward requires a strong heuristic baseline to be effective, which could limit portability across problem domains without such heuristics.

---

> ### Author Rebuttal · Authors · 2025-07-31
>
> Thank you for your thoughtful evaluation of our manuscript and the valuable feedback. We are glad that the strengths of our paper were recognized. In this rebuttal, we respond to your concerns and clarify several aspects of our work.
>
> #### Weakness 1:
> >The method underperforms or matches baselines on simpler DAG structures (e.g., TPC-H). While mentioned, this is not deeply analyzed.
>
> We analyze this issue as follows:
> Our proposed CAI intends to remove the redundant priority comparison among DAG nodes that has dependency relationship, thus obtaining more accurate sampling probability. So, CAI is particularly useful in DAGs with more complex dependency structures, in which multiple nodes often lie along a longer path. For simpler DAG structures (e.g., TPC-H benchmark), they typically consist of many small-size DAGs with much shorter paths. In this situation, the number of dependent nodes located on the same path is small, and the existing dependencies are weak. The redundant comparisons among dependent nodes are significantly reduced, so the further improvement from applying CAI is limited. We thank the reviewer for pointing out this to improve our analysis about the experiment.
>
>
> #### Weakness 2:
> >Like most one-shot methods, it sacrifices interpretability compared to MDP-based incremental approaches.
>
> Most one-shot DAG scheduling methods are generally less interpretable than MDP-based incremental approaches. This is because there is no intermediate feedback to help these methods establish the connection between each decision step and the final outcome. However, by introducing the dense reward signal, our method improves the interpretability of one-shot approaches to some extent. Specifically, the nodewise decisions, sampling probabilities and dense reward signals can be regarded as an entire MDP sampling trajectory in RL (like a trajectory by Monte Carlo sampling). This makes the interpretability of one-shot learning closer to MDP-based incremental approaches than those one-shot methods with sparse rewards.
>
> #### Weakness 3:
> >Dense reward requires a strong heuristic baseline to be effective, which could limit portability across problem domains without such heuristics.
>
> We conducted experiments comparing multiple heuristic advantage baselines, including the previously used HEFT, and Critical Path on a Processor (CPOP), Shortest Finish Time (SFT). The results of the gap related to HEFT in percentage is shown in the following table, as a complement to Table 1. The results show that our method with different advantage baseline still outperforms existing methods, and the performance of our method with different heuristic advantage baseline is comparable.
> |Method|SIPHT100|SIPHT200|SIPHT300|SIPHT400|
> |----| --| ----|----|---|
> |ours+HEFT|-13.3|-5.42|-0.25|-0.62|
> |ours+CPOP|-12.2|-5.42|-0.20|-0.62|
> |ours+SFT |-12.8|-4.83|-0.22|-0.62|
>
>
> #### Question 1:
> > What are the disadvantages of relying on Q-Learning for the generation of a dense reward? Could a simple feedback per scheduled task performance be used to make the reward dense for each scheduling step?
>
> We’d like to clarify that our method is based on policy gradient, rather than Q-learning. This design is motivated by the nature of the RL-based one-shot scheduling, which relies on differentiable probabilistic modeling of nodewise scheduling decisions. Such probabilistic modeling can be achieved in policy gradient. However, standard Q-learning typically does not explicitly model the action probability.
>
> We think the reviewer might suggest that we could fit a Q-function network to estimate per-node scheduling values (or rewards) to provide dense feedback signal. However, in practice, in each DAG scheduling problem instance, we only obtain a single global performance metric, making the per-node value fitting process inherently sparse. As a result, introducing such additional fitting is unlikely to be very effective.
>
> In contrast, our designed dense reward signal reflects the accumulated impact of the scheduling trajectory from the beginning up to the current step. The reward approximates the overall performance of the current schedule. In this sense, our dense reward signal can make a close advantage function estimation over the current execution state of the DAG. So, fitting a Q-function network is unnecessary.
>
>
> #### Question 2:
> >How sensitive is the method’s performance to the choice of the heuristic used for advantage baseline estimation?
>
> As we clarified in the response to Weakness 3, the performance our our method with different heuristic advantage baseline is comparable.
>
> #### Question 3:
> >Are there theoretical guarantees (e.g., bounds) on performance improvements or convergence properties?
>
> Since the optimization of our approach is based on policy gradient, it inherits the theoretical convergence properties of the classical policy gradient framework. Although the convergence performance in actual usage might be not as perfectly good as that of the classic policy gradient under ideal conditions, several designs in our method could enhance the stability and convergence behavior in practice:
> 1. Nodewise dense rewards signal provides a closer approximation of value/advantage estimation at each decision step.
> 2. The CAI module improves the quality of nodewise decision probability estimation, obtaining more accurate gradients and hence improved convergence behavior.
> 3. For a given DAG scheduling instance, the results generated by the heuristic advantage baseline algorithm is fixed across training epochs. This ensures that the advantage estimation is unbiased.
> While these designs do not provide formal theoretical bounds, they provide empirical support for training stability and convergence, as reflected in the performance across diverse testbenches.
>
>
> #### Limitation 1
> >The method assumes access to a decent heuristic for baseline reward estimation.
>
> Please check our Response to Weakness 3.
>
> #### Limitation 2
> >CAI's benefit is less pronounced on shallow or disjoint DAGs.
>
> Please check our response to Weakness 1.
>
> #### Limitation 3
> >Despite improved training efficiency, the GNN-based architecture and sorting overhead may still limit real-time or very large-scale applications.
>
> We would like to highlight that our method inherits the runtime advantages of one-shot scheduling compared to some other RL-based approaches, as demonstrated in the results. This is because one-shot scheduling avoids repeated encoding of the entire workflow, which offer runtime advantages in certain large-scale scenarios. This suggest that our method have potential for further extension and optimization in large-scale environments, like scientific workflow management, which we consider a valuable future extension direction.
>
>
> #### Limitation 4
> >The method does not directly model dynamic environments or online scheduling, restricting its scope to static DAG instances.
>
> While our proposed one-shot policy is evaluated in relatively static settings, it can be applied to dynamic environments. For example, when new tasks are revealed, or a new DAG job instance is submitted, re-scheduling can be performed by re-running the network. The objective and dense reward signal calculation should be adjusted accordingly. In cases where DAG job structures and arrival time is known in advance, we can perform pre-processing by adding pseudo predecessor nodes with fixed execution times.
>
> Nevertheless, real-world dynamic scheduling scenarios often involve domain-specific constraints, which might require more dedicated modeling. Therefore, we agree this is a promising future work.
>
>
> #### Paper Formatting Concerns
> >Limitations section discussing the model limitations is not clearly stated. Having a distinct paragraph or subsection would improve the quality of the paper.
>
> Due to the page limitation, we only discussed limitations in Appendix G in the supplementary material. However, in order to improve the paper quality, we fully agree the viewer’s valuable suggestion to use a distinct subsection in the main text to state the limitations in the camera-ready version. Moreover, we will improve the limitations discussion based on the points that the viewer has raised.

---

> > ### Comment · Reviewer_ebqq · 2025-08-05
> >
> > I thank the authors for the clarifications.
> >
> > For the Q-Learning question, I was referring to a potential of using an Actor Critic framework, such as SAC, where the Critic learns to provide a dense feedback to the actor.
> >
> > After reading the rebuttal and the clarifications to the questions. I would keep this score.

---

### Decision · Program_Chairs · 2025-09-17

**Decision:**

Accept (poster)

**Comment:**

The paper proposes a novel one-shot DAG scheduling method using reinforcement learning (RL), and also propose a dense reward signal for node level decision-making optimization in training, effectively addressing the sparse reward challenge.

Based on comments from reviewers, the strengths of the paper include: (1) Strong empirical results: The proposed method outperforms existing learning-based DAG scheduling approaches across multiple heterogeneous benchmarks. The evaluation is thorough, including ablation studies that highlight the contribution of each component. (2) Clear theoretical foundation: The paper presents a well-structured analysis of limitations in prior work and supports the proposed method with formal definitions, lemmas, and sound justification.

Some minor concerns were raised, such as the high computational load due to data dependency and per-round adjustability, as well as the limited exploration of alternative or multi-objective metrics, which may affect general applicability. However, these are relatively minor and do not undermine the overall quality or contribution of the work.

Given the overall positive reviewer scores—one Accept and three Weak Accepts—and the paper’s solid technical merit, I recommend acceptance.